# Influence of Atmospheric Pollutants on Allergic Sensitization to Cupressaceae, *Olea*, and *Platanus* Pollen in the Community of Madrid (2017–2021)

**DOI:** 10.3390/life15111774

**Published:** 2025-11-19

**Authors:** Javier Chico-Fernández, Angélica Feliu Vila, Beatriz Rodríguez-Jiménez, Teresa Valbuena Garrido, Esperanza Ayuga-Téllez

**Affiliations:** 1Programa de Doctorado Ingeniería y Gestión del Medio Natural, ETSI de Montes, Forestal y del Medio Natural, Universidad Politécnica de Madrid, 28040 Madrid, Spain; 2Servicio de Alergología, Hospital Universitario del Tajo, 28300 Madrid, Spain; angelica.feliuvila@salud.madrid.org; 3Servicio de Alergología, Hospital Universitario de Getafe, 28905 Madrid, Spain; brodriguezj@salud.madrid.org; 4Sección Alergología, Hospital Universitario Infanta Sofía, FIIB HUIS-HUHEN, 28702 Madrid, Spain; mteresa.valbuena@salud.madrid.org; 5Buildings, Infrastructures and Projects for Rural and Environmental Engineering (BIPREE), Universidad Politécnica de Madrid, 28040 Madrid, Spain

**Keywords:** tree pollen, air pollution, pollen sensitization, pollinosis

## Abstract

Tree pollen is the most abundant in the Community of Madrid (CAM), and specifically, pollen types from *Olea*, Cupressaceae, and *Platanus* are the most allergenic, after Gramineae, in this Spanish region. Air pollutants are one of the most significant stress factors for wind-pollinated vegetation, especially in urban areas, and can cause alterations in the immune system and the consequent triggering of type I hypersensitivity reactions mediated by immunoglobulin E (IgE). This study analyses the allergic sensitization caused by the interrelation of O_3_, NO_2_, and PM_10_ pollutants with the tree pollen types *Olea*, Cupressaceae, and *Platanus* in the period 2017–2021. To this end, general linear models were calculated using the Statgraphics Centurion 19 tool. The data collected came from the Air Quality Networks of the CAM and Madrid City Council, the CAM Palynological Network, and the Allergy Services of the reference hospitals in the five study areas. This research confirms a statistically significant correlation between allergic sensitivity to pollen types and their concentrations in the air, and those of atmospheric pollutants, in the different areas and years studied. These pollen and pollutant concentrations in the atmosphere of the CAM jointly influence the prevalence of allergic sensitisation, as is evident in all the models calculated.

## 1. Introduction

Of all possible sources of airborne allergens, pollen is the most significant and the leading cause of sensitisation worldwide. Furthermore, in recent decades, it has been the main cause of asthma and allergic or extrinsic rhinoconjunctivitis in the Community of Madrid (CAM), as well as in Europe (where these respiratory allergic diseases affect between 15 and 40% of the population) and, by extension, worldwide [1,2,3,4].

Airborne pollen can cause immediate reactions, i.e., IgE-mediated type I hypersensitivity, which occur in two phases, sensitisation and effector, in patients with respiratory allergic diseases such as allergic rhinitis, allergic rhinoconjunctivitis and asthma, as well as other conditions such as atopic dermatitis, acute urticaria/angioedema, and allergies to food, insect venom and medicines [4,5].

The sensitisation phase involves a complex interaction between the adaptive and innate immune systems and depends on signals from T2 cells, which regulate the production of allergen-specific immunoglobulin E (sIgE). This process begins when a person is exposed to an allergen for the first time, and it is taken up by antigen-presenting cells (APCs), such as macrophages (Mφ), B lymphocytes, and dendritic cells, which process and present the allergen peptides on their surface [5].

In the effector phase, mast cells and basophils become sensitised to the allergen and are coated with sIgE antibodies. This ultimately leads to the release of mediators such as histamine, as well as cytokines and chemokines, causing the characteristic symptoms of an allergic reaction, i.e., inflammation of the affected epithelial tissue or mucosa, whether bronchial, nasal, conjunctival, or other. In addition, symptoms include vasodilation, smooth muscle contraction, increased vascular permeability, as well as stimulation of sensory nerves and mucus production [5].

In the CAM, and, by extension, in the Mediterranean region, airborne *Olea* pollen is the second most common cause of allergy, after Gramineae, among the population sensitised to pollen aeroallergens. This is due to the abundance of olive trees (*Olea europaea*) and the allergenic potential of this type of pollen [1].

In addition, the extensive planting in urban areas of notably allergenic species from two other tree taxa, Cupressaceae and *Platanus*, means that people in these areas are increasingly exposed to the pollen that these trees have been released into the atmosphere in recent decades [1].

Thus, these three types of tree pollen are among the most abundant in the cities of the study region, along with those of *Quercus*, *Pinus* and others, and have experienced an increase in their concentration in the air that has been documented from 1994 to 2022 [2].

The third and fourth most common causes of allergies in the CAM are pollen from Cupressaceae and *Platanus*, in that order within Cupressaceae, the most relevant allergenic species are *Cupressus arizonica* and *Cupressus sempervirens* [6,7], although *Cupressus arizonica* is considered to be the source of the main concentrations of Cupressaceae pollen in the atmosphere of the CAM, given that its flowering coincides with the highest incidence, and also due to its abundance [8].

In contrast, dust mites are only the source of allergies in a small percentage of patients in the CAM, due to the continental climate of this geographical area. However, in areas with a Mediterranean climate such as Barcelona, dust mites are the most common environmental agents causing allergies, ahead of pollen. And, among the types of pollen, *Platanus* is the main cause of pollen allergies in this coastal city in north-eastern Spain [6].

The Madrid region is exposed to one of the highest concentrations of *Platanus hispanica* pollen in the entire Iberian Peninsula [6]. Furthermore, it is the most abundant species in the streets and green areas of 14 of the 21 districts of the capital [9], and specifically the second most abundant, after *Styphnolobium japonicum* in the Salamanca neighbourhood, a district that represents one of the five areas in this study.

Pollination usually begins in the second half of March, is highly abundant and typically concentrated between two and four weeks, and experiences interannual variations due to changes in flowering intensity caused by the severe pruning that these trees usually undergo, as well as changing weather conditions [8]. However, compared to other pollens that reach lower atmospheric concentrations, such as those of *Olea* and Gramineae, *Platanus hispanica* pollen has a lower sensitisation frequency [6].

Nevertheless, not only has the pollen concentration of *Platanus* been increasing in recent years, since it was first considered an important source of allergy in the CAM in 1997, but this type of pollen has also been revealed as one of the most significant causes of pollinosis in the Madrid region [6]. Patients allergic to this type of pollen who live in the CAM are mainly sensitised to the allergen Pla a 2 (ahead of Pla a 1, which together with Pla a 2 is usually the most common allergen in *Platanus*), a 43 kDa glycoprotein with polygalacturonase activity, which is detected in 84% of patients [6].

A total of 48.7% of patients allergic to *Platanus* pollen in the CAM suffer from asthma, a percentage that falls within the range of values reported for pollen-allergic patients in Spain, considering the frequency of asthma in pollen-allergic patients with food allergies (59%) or without food allergies (47%) [6]. Furthermore, most of these patients allergic to *Platanus* pollen (89.5%) are sensitised to other types of pollen [6].

*Olea* pollen begins to be recorded at the Palinocam Network stations in April, until May or June, with average daily concentrations not exceeding 100 g/m^3^ except for a few days of the year. It comes from *Olea europaea* and is considered one of the main causes of pollinosis in the CAM [8].

Ole e 1 is the most relevant allergen of the *Olea* pollen type, as it sensitises at least 70% of Spanish patients and is the main cause of rhinoconjunctival and bronchial symptoms associated with allergy to this pollen. It has a high sequence identity with the 20 kDa acidic glycoproteins of pollen from the *Olea*ceae family [7,10].

It has been found that patients sensitised to Cupresaceae pollen often experience symptoms of rhinoconjunctivitis and, in recent years, asthma [11], although the former condition is more common, and this is also common with *Olea* pollen [7]. Likewise, polysensitisation to both types of pollen is very common [7].

Allergies to Cupressaceae and *Olea* pollen are becoming increasingly important in Spain, with differences in the prevalence of sensitisation depending on the geographical area [7].

The Cupressaceae family consists of species that are resistant to air pollution, and their pollen is the main cause of respiratory allergic diseases in winter, especially in North America, Japan and Mediterranean countries such as Spain, particularly in the central region. It is practically the only allergenic pollen present in the atmosphere of Madrid during the cold season [8,11,12].

Its pollen season extends from autumn to April of the following year [8]. The main allergens of the most allergenic species, namely *Cupressus arizonica* and *Cupressus sempervirens*, are Cup a 1 and Cup s 1, which are pectate lyases, both of which have a sequence identity of greater than 95% [7].

Cupressaceae pollen ranks second in the CAM, behind *Platanus*, in terms of its atmospheric incidence, as it can reach 16.1% of the total annual pollen in that region [8]. In fact, the CAM can be defined as a geographical area with high exposure to this type of pollen, especially in residential areas further away from urban centres, where the planting of isolated specimens, or to form hedges, has been widespread at least in recent decades. This leads to an increase in peak pollen concentration values in the atmosphere and, an increase in the percentage of sensitisation, especially in the more industrialised areas of the south of the region. In these areas, the emission of polluting particles is higher. This, in turn, implies an interaction of these atmospheric pollutants with pollen grains, resulting in a greater availability of pollen allergens in the atmosphere and, in turn, an increase in allergenicity [7,11].

Air pollution, considered in isolation, is already a significant public health problem due to its involvement in the increasingly evident rise in the prevalence of allergic diseases, including asthma [4,13].

In addition, atmospheric pollutants, together with meteorological factors, are two of the most important stressors for vegetation, in general, and particularly for wind-pollinated trees. Especially those located in urban areas, which can cause alterations in the concentration of allergenic bioparticles such as pollen grains in the air, with the consequent release of allergenic proteins and biogenic adjuvants [4,14,15].

Air pollutants can also alter the immune system and human physiological processes, specifically by disrupting protective epithelial barriers, thereby increasing permeability and causing damage to the epithelium [4,14,15]. This facilitates the access of these pollutants to the submucosa, where they can interact with the smooth muscle cells of the airways and fibroblasts, causing processes such as inflammation, nitrosative and oxidative stress, and alteration of the microbiomes, which can lead to allergic diseases [4,14,15].

In addition, these pollutants inhibit ciliary beat frequency, resulting in delayed clearance of inhaled aeroallergens and irritants [4,15].

As adjuvants, atmospheric pollutants can also bind to allergenic proteins in airborne allergens, stimulating IgE-mediated responses, with the consequent chemical modification and alteration of their immunogenicity, thereby causing the exacerbation of the manifestation of various allergens contained in pollen grains, as well as the stimulation of immune responses in affected individuals [4,12,15].

In addition, these agents cause the alteration and degradation of the exine, with the consequent dispersion of pollen allergens into smaller fractions (such as cytoplasmic pollen grains), the alteration of their allergenicity, and an increase in their bioavailability, as they gain the ability to penetrate more deeply into the respiratory tract [4,15,16,17,18].

Air pollutants, with the additional influence of meteorological factors, can influence the release not only of allergenic pollen proteins and fungal spores, but also of pollen-associated lipid mediators (PALMs). These act as adjuvants, and although they are bioactive, they are not allergenic on their own, which does not prevent them from causing immunomodulatory and pro-inflammatory effects and triggering and exacerbating allergies [15].

Thus, skin prick tests with pollen allergens can cause larger wheals when performed with low-molecular-weight compounds extracted from pollen. PALMs can be found in much higher concentrations in pollen collected near busy roads. Phytoprostane PALMs can promote a Th2 T-cell allergic response by inhibiting the production of interleukin 12 (IL-12) by dendritic cells in the lower respiratory tract. Leukotriene-like PALMs can attract and activate innate immune cells such as eosinophils and neutrophils [15].

Furthermore, the adhesion of pollen allergens and adjuvants to the surfaces of contaminating particles (sorption layers, protein crowns) facilitates their transport to peripheral and deep pathways and could enhance sensitisation and allergic response by providing multiple/multivalent epitopes that facilitate receptor cross-linking [15].

Asthma patients may experience exacerbated symptoms because of atmospheric pollutants binding to pollen grains, leading to increased susceptibility to exposure to airborne allergens. When this binding occurs specifically with particles (PM_2.5_ and PM_10_), it can alter the characteristics of pollen allergens [13,14,19]. These particles, together with SO_2_, are associated with increased respiratory distress in children [4,20]. In addition, O_3_ and NO_2_, together with CO, increase the likelihood of asthma in babies with bronchiolitis, and exposure to NO_2_, together with CO, is associated with a higher prevalence of childhood asthma [4,20].

Nitrogen oxides, particulate matter (PM), and O_3_, when acting as adjuvants, together with other pollutants and chemicals, mainly anthropogenic, can generate various pro-allergic effects [15].

NO_2_ originates mainly from human activity, specifically from all types of high-temperature combustion processes, such as those triggered in internal combustion engines, which are the main emitters of this primary pollutant. It is also generated in conventional thermal and cogeneration electricity generation processes, as well as in industrial combustion plants and production processes and in waste treatment and disposal plants [21]. NO_2_, in addition to NO, causes nitration of allergens, as well as a tendency towards a Th2 response, increased eosinophilic inflammation and airway permeability [15].

PM_10_ particles, or particles with a diameter of less than 10 µm, remain suspended in the air and, although they are not as harmful as PM_2.5_, they can be deposited in the periphery of the lung, an area particularly prone to injury, from which they are very difficult to remove [21].

Particulate matter (PM) facilitates the production of reactive oxygen species (ROS) and induces inflammatory processes. In addition, they increase the expression of co-stimulatory molecules in dendritic cells (MHC class II, CD40, CD80, CD86), the main antigen-presenting cells, which act as sentinels of the immune system [15].

Generally, their origin is mainly natural, and less than 20% of the particles are generated because of combustion processes, in addition to losses generated in industrial processes such as mining, quarrying, cement production, rock crushing, and solid urban waste treatment, among others [21].

However, in large cities such as Madrid, almost half of the mass of PM_10_ particles contained in the atmosphere may correspond to road traffic. Meanwhile, the contributions of cortical and secondary aerosols may be similar and significant in this city. The presence of marine particles is insignificant due to the distance from the coast [22].

However, the Atlantic Ocean is an intermediate source of sodium (Na), which implies a significant influence of remote natural sources of PM_10_ in Madrid. The Sahara Desert is identified as the source of PM_10_ particles of cortical origin on days when maximum concentrations of these particles occur. The route followed by Saharan dust plumes reaching the city of Madrid from the Sahel region is also identified [22].

O_3_ is a secondary pollutant, meaning that it is not generated directly in the atmosphere, but because of interactions between primary pollutants (specifically because of the photolytic cycle of nitrogen oxides, involving NO and NO_2_), natural components of the atmosphere, and incident radiation. At sufficient concentration, it can cause, among other effects, irritation of the respiratory system, inflammatory lung injury with particularly intense effects in people with respiratory diseases such as asthma and Chronic Obstructive Pulmonary Disease (COPD). Even at very high concentrations, it can be lethal [21,23].

O_3_ also generates oxidative stress, in addition to inflammation and increased airway permeability. It also causes the formation of protein ROS (reactive oxygen species) and protein dimers, with elevated levels of the complement protein C3a [15].

This study hypothesises whether pollen and pollutant concentrations in the atmosphere of the CAM are statistically significantly associated with allergic sensitisation to the three pollen types studied in the population of the Madrid region. This is a pioneering study in the CAM, as until now these interrelationships between the concentrations of both types of atmospheric agents and the prevalence of pollen sensitisation had not been analysed in this region of Spain.

## 2. Materials

This study focuses on the years 2017, 2018, 2019, 2020 and 2021. To carry it out, data has been collected on the concentration of pollen in the atmosphere from three types of tree pollen that are among the most allergenic in the CAM (Cupressaceae, *Olea* and *Platanus*) [1]. In addition, concentration data has been provided for three atmospheric pollutants, which are among the most harmful to human health (O_3_, NO_2_ and PM_10_) and are measured at air quality stations in the five study areas. Finally, data on allergic sensitisation to the pollen types under study has also been obtained.

The pollen data has been provided by the CAM’s Palinological Network (Red Palinocam), part of the CAM’s Department of Health. The air pollutant data has been provided by the Air Quality Network of the Regional Ministry of the Environment, Land Planning and Sustainability of the CAM, as well as by the Air Quality Network of Madrid City Council.

These two types of data are daily and have been obtained from measurements taken by sampling stations. They have been described in detail in a previous study, which analysed the correlation between both factors [19].

That study analysed a total of six types of tree pollen and six atmospheric pollutants, including those present in this research. In addition, the study included the mean values, standard deviation and coefficient of variation for both types of independent variables. In the case of pollen types, their percentage representation with respect to the total pollen collected at the 11 stations of the Palinocam Network is specified [19].

The geographical location and nomenclature of the measuring stations for the two types of variables are also indicated, along with a description of tree biodiversity and an analysis of the benefits of its presence in the CAM [19].

The sensitisation data, also collected daily, have been provided by five public hospitals in the CAM, located in the five areas of influence of the Palinocam Network stations, as shown in Table 1. This information has been provided by the respective heads of the Allergology departments, with the authorisation of the directors of these centres, as well as the acceptance of the study by two Research Ethics Committees from the Getafe University Hospital and the Gregorio Marañón University General Hospital (CEIm 23/68, approved on 25 January 2024, and CEIM 23/68, approved on 4 March 2024, respectively).

The data provided by the allergy laboratories of the hospitals participating in this study consist of positive results for the three types of pollen studied, determined using skin prick tests, in the case of the Tajo University Hospital and the Severo Ochoa University Hospital. The other three hospitals have provided the sIgE results obtained through blood tests.

It should be noted that in the Madrid area: Barrio Salamanca, the Gregorio Marañón University General Hospital was selected as the closest possible to the corresponding pollen station and as the one that could provide the necessary data for this study. In the case of the other four areas, this information has been obtained from the specific hospitals located near the pollen concentration measurement stations and, therefore, from the health centres listed in Table 2.

### 2.1. Allergy Sensitisation Data

Both asthma patients and those suffering from allergic rhinoconjunctivitis in the CAM are mainly sensitised to Gramineae, *Olea*, Cupressaceae and *Platanus* pollen types. Furthermore, the percentage of allergic sensitisation to these types of pollen has increased in this region of Spain between the studies reflected in the 2005 and 2015 editions of Alergológica [3]. (A more detailed explanation is included in the Appendix A).

### 2.2. Tree Data

The urban tree inventory data from the five study areas indicate that, except in Getafe (where Cupressaceae trees are the most abundant), the highest presence of the three tree taxa is *Platanus*, followed by Cupressaceae and *Olea*. Furthermore, the highest proportion of *Olea* and *Platanus* trees in relation to the total number of specimens is found in Aranjuez, followed by Getafe and Leganés, respectively. Alcobendas has the highest proportion of Cupressaceae trees, ahead of Getafe (as can be seen in Appendix A).

Furthermore, as can be seen in Figure 1, the municipality of Leganés has the highest tree density, followed by Aranjuez. In other words, these two study areas are home to the highest number of trees in relation to strictly urban areas [24,25,26,27,28,29,30,31].

## 3. Methods

The data provided for the independent variables (pollen concentration and concentration of pollutants in the atmosphere of the CAM) are daily. The same applies to the dependent variable (pollen sensitisation). These three data sets were obtained on the same dates, which makes it possible to establish an adequate correlation between them, defined by the coefficients of determination. Based on these daily data, monthly average values have been calculated for each year and each of the five geographical areas studied.

The degree of relationship between the dependent variable and each of the independent variables was established by calculating general linear statistical models using the Statgraphics Centurion 19 data analysis tool.

In each of the equations obtained, the concentration of the pollen type corresponding to the dependent variable, as well as the concentrations of the three atmospheric pollutants under study, have been introduced as independent variables (quantitative factors). Thus, for example, in the model calculated for sensitisation to Cupressaceae, the concentration of this pollen type is included as an independent variable, in addition to the concentration of O_3_, NO_2_ and PM_10_. Likewise, the coefficients relating to the populations and the years of study have been included, as can be seen in Table A1.

Therefore, for the calculation of the models, each of the five study stations and each of the five years of the 2017–2021 period have been included as categorical or qualitative factors.

Two options are included as possible dependent variables that aim to express and define patient sensitivity to Cupressaceae, *Olea* and *Platanus* pollen in the calculations performed. The first consists of the ratio between the number of positives and the total number of patients undergoing sensitivity testing (P/T) in each of the months in which these tests were performed during the study period.

The second option is defined as the ratio between the number of positives and the total population of each of the five study areas (P/PobT). This population has been obtained as the result of the sum of the respective total assigned populations, i.e., of all ages (collected by the Madrid Health Service Results Observatory in the 8th, 9th and 10th Primary Care Reports [32]), attended by each of the health centres included in the five study areas of influence of the Palinocam Network stations. These populations are included, together with the names of the health centres, in Table 2.

It was initially understood that either of the two options mentioned above could be valid for defining patient sensitivity to pollen from Cupressaceae, *Olea*, and *Platanus*, the three tree pollen taxa that are among the most abundant in the air in the CAM, according to various studies [1,2]. However, a comparative statistical analysis was also performed between the two options to confirm this initial impression.

The *p* values corresponding to Fisher’s F ratio statistic have been calculated for both the calculated models, reflected in the results of the analysis of variance for those models and the *p* values to assess whether the coefficients of each of the independent variables included in each of the calculations, as well as the categorical factors (population and year), are equal to zero.

In all the models calculated, standardised residuals greater than 3 standard deviations have been generated. The rows of data corresponding to these residuals have been eliminated, and the model has been recalculated as many times as new possible outliers have arisen that could distort the resulting model. In none of the cases this resulted in the loss of information from any of the independent variables, but in all of them, the adjustments were improved with respect to the initial calculations, and therefore higher adjusted R^2^ values were obtained.

In addition to the elimination of outliers, in order to validate the calculated models, the ANOVA test of the model, the value of the coefficient of determination and the diagnosis of residuals using the Kolmogorov–Smirnov goodness-of-fit test to the normal distribution were used, on the basis of which it is possible to check the extent to which the distribution of these residuals resembles a normal distribution.

Fisher’s F-test was also used for this validation, to check for significant differences in the mean values of the residues between the different years of study. Finally, Levene’s test was used to verify the homogeneity of the variances, i.e., to ascertain whether there are statistically significant differences between the standard deviations of the residuals, with the chosen confidence level, which, in all the tests mentioned, was 95%, so that the significance level adopted in all of them for testing the hypotheses was 5%.

## 4. Results

Table 3 shows the mean values, standard deviation and coefficient of variation in the independent variables included in this study. As can be seen, the highest mean value among the pollen types is that corresponding to *Platanus*, with 47.1563 pollen grains per m^3^ (g/m^3^). This type of pollen also has the highest standard deviation, at 159.515 g/m^3^, which is why the coefficient of variation is also very prominent (338.269%), and very similar to that obtained for *Olea* (339.339%), because the standard deviation value (46.4434 g/m^3^) is significantly higher than its mean value (13.6864 g/m^3^), which, on the other hand, is the lowest of the three pollen types studied. Cupressaceae ranks between *Platanus* and *Olea* in terms of mean values, with 32.2254 g/m^3^. However, despite having a comparatively much lower coefficient of variation (203.304%), this is still very high, as its standard deviation (65.5155 g/m^3^) is double its mean value.

In contrast to pollen concentration values, the coefficients of variation for the three atmospheric pollutants studied are much lower, as their standard deviations are significantly lower than their mean values (Table 3). Among the latter, the most notable is O_3_, with 52.0809 µg/m^3^, followed by NO_2_, with 31.4869 µg/m^3^, and PM_10_, with 19.7984 µg/m^3^.

On the other hand, it may be useful to analyse the relationship between pollen concentrations and the two types of dependent variables, P/T and P/PobT, already discussed in the Methods section, and to try to observe the differences existing in the five study areas within the period 2017–2021. To this end, the pollen type *Olea* has been selected, which is the most allergenic of the three analysed in this study, as argued in the Introduction, while also being the least represented in terms of *Olea europaea* specimens in all the geographical areas analysed.

Therefore, Figure 2 shows graphs illustrating the relationships between the atmospheric concentration of *Olea* and the sensitisation variables, P/T and P/PobT, in relation to this type of pollen.

As can be seen, the five study areas show the most relevant pollen concentrations (sky blue areas), centred approximately between April and June of each of the five years of the study. However, for example, in all seasons in 2019, this pollination was much less abundant than in the other years. This is also strikingly the case in 2020 in Aranjuez (Figure 2).

Furthermore, the highest peaks in *Olea* pollen concentration occur, in this order, in Getafe, Aranjuez and Leganés, which coincides with the number of *Olea europaea* trees inventoried, as can be seen in Appendix A. In fact, the lowest peaks of *Olea* pollen occur in Alcobendas, which has 364 olive trees, and Barrio de Salamanca in Madrid (the station where pollen concentrations reach a maximum of 140 g/m^3^, as can be seen in Figure 2), with only 44 specimens.

On the other hand, allergic sensitisation to *Olea*, expressed by the P/T variable (green line), reaches its highest amplitude at the Getafe station, where values ranging from 0.00 to 0.90 are found. This zero value in Getafe is the only one found in all the study areas, and coincides with the start of pollination in 2020, as can be seen in Figure 2. The Leganés station is the most like its neighbour, Getafe, in terms of the range of values of the P/T variable for *Olea*, as it reaches a similar maximum value, although, in principle, it does not reach zero values. However, the lack of sensitisation data in Leganés means that the line is interrupted in 2017 and at the beginning of pollination in 2020.

Aranjuez and Madrid Barrio de Salamanca adopt P/T values with a very similar range (between 0.30 and 0.70), while Alcobendas concentrates its values between 0.60 and 0.80, therefore with the lowest amplitude in the comparison.

In general, in the five stations, the peak values of the P/T curve for *Olea* occur immediately after, i.e., in the months following, the pollination periods, and after this rise, there is a downward trend in these values in the following months.

Finally, the values of the P/PobT variable for *Olea* are restricted to a much narrower range than in the case of P/T. This is even though they have had to be multiplied by 200 to analyse them using the purple curves that appear in the five stations (Figure 2).

In this case, the greatest variations in values occur in Alcobendas, with a range of approximately 0.05 to 0.45. Aranjuez is the second study area with the widest range of values (approximately 0.00 to 0.25). In contrast, the other three study areas do not obtain P/PobT values for *Olea* greater than 0.10, which translates into slightly broken lines with hardly any noticeable ups and downs, as can be seen in Figure 2.

A certain parallelism can be seen between the curves obtained for each type of sensitisation variable (more noticeable in the Alcobendas station and more discreet in the Aranjuez station) if the different months and years of the study period are visually scanned. Therefore, in the case of P/PobT for *Olea*, a tendency for its values to rise after the pollination periods can also be seen, although less clearly than in the case of P/T. This is less noticeable in the case of the Madrid Barrio de Salamanca, Getafe and Leganés stations (Figure 2).

### Results Obtained with General Linear Statistical Models

Based on the calculations made using general linear statistical models, the values for the 2017–2021 study period are shown in Table 4, and the corresponding equations are listed in Table A1 in the Appendix B.

Each of the calculated models includes: the dependent variable, four quantitative factors (one of which is the pollen type related to the dependent variable, and the other three are the atmospheric pollutants under study, i.e., O_3_, NO_2_ and PM_10_) and two categorical factors, namely season and year.

Based on these results (Table 4), it can be stated that:In the calculated models, *p* values lower than 0.05, which is the chosen significance level, were obtained with a confidence level of 95%. In fact, in all of them, the *p* values were equal to 0.000, which implies that all the models are explanatory, i.e., there is a statistically significant relationship between the independent variables and the dependent variable in every one of the calculations performed.The lowest adjusted R^2^ value is 54.5993%, which is obtained for *Olea* and P/T. In all other cases, the coefficient of determination exceeds 81%.Most of the *p* values calculated for the coefficients of the quantitative independent variables (atmospheric concentration of pollen from the three pollen types and air concentration of the three atmospheric pollutants studied) imply acceptance of the null hypothesis that the model coefficients are equal to zero. However, the results reflected in the previous points allow us to consider that the models are significant and explain the dependent variable with a good degree of fit (percentage of variability explained by the medium–high model).

As can be seen in Table 4, the number of observations in the six calculations performed is, in all cases, different from 300 (which is the maximum number of monthly observations that can occur, taking into account the five seasons and the five years of study), due, firstly, to the elimination of data rows relating to atypical residues. Secondly, it is due to the lack of data on pollen types, pollen concentration, or some of the atmospheric pollutants. Thus, in the case of the Leganés station, there is no data available on sensitisation for the first five months of 2017 or for April 2020. In April 2020, the Alcobendas and Getafe stations were unable to count pollen grains for any of the three pollen types studied, as was the case in April and May 2020 at the Aranjuez station. The same occurred in August 2019 at the Madrid Barrio de Salamanca station. Finally, in January 2020, no data on O_3_ and NO_2_ concentrations were recorded at the Alcobendas station.

The *p*-value of the models obtained for P/T and P/PobT, both for Cupressaceae and for *Olea* and *Platanus*, is less than 0.05, specifically 0.0000 in all six cases (Table 4); therefore, in all of them, there is a statistically significant relationship between these dependent variables and the predictor variables, with a confidence level of 95%.

As for the two categorical factors involved in the models, the *p* value is also less than 0.05 in all of them except in the case of the year factor, for *Olea* and P/T, with a *p* value of 0.2854 (Table 4).

However, in the six models calculated, the *p* value for pollen types is greater than 0.05. Therefore, these terms in isolation are not statistically significant, i.e., they do not individually explain the variability of P/T or P/PobT (Table 4).

Regarding atmospheric pollutants, only in the case of the adjustments made for P/T and Cupressaceae are the *p* values for O_3_, NO_2_ and PM_10_ less than 0.05 (specifically 0.0002, 0.0114 and 0.0021, respectively). This also occurs in the case of P/T with *Platanus*, with a *p* value of 0.0054 for O_3_.

Therefore, in all calculated models, at least one of the quantitative factors (pollen types and atmospheric pollutants) cannot alone explain the variability of the dependent variable (allergic sensitisation). However, taken together, these factors can explain this variability, with coefficients of determination, adjusted R^2^, of at least 54.5993% (in the case of P/T with *Olea*). This means that all the calculated models can explain the variability of the dependent variable in a very significant way (Table 4).

On the other hand, in the six models calculated, there are initially (since the data rows have been repeatedly eliminated, with the models being recalculated until the atypical residuals are completely eliminated) between three and seven studentised residuals greater than three standard deviations. It has been verified that the improvement in the coefficient of determination resulting from the elimination of the data corresponding to these outliers from the models contributes to an improvement in the fit of at least 2.38% (in the case of P/T for Cupressaceae), and that in the case of P/T for *Olea*, it is greater than 10% (since before and after the elimination of these data, the adjusted R^2^ value goes from 44.3127% to 54.5993%), so these rows of data have been eliminated from the calculations.

The best fits are for P/PobT, in the case of Cupressaceae and *Olea*. The most striking case is that of *Olea*, since, as can be seen in Table 4, the difference in the adjusted R^2^ results is greater than 37% with respect to P/T (92.1093% versus 54.5993%). However, in the case of Cupressaceae, the *p* values for O_3_, NO_2_, and PM_10_ are less than 0.05 (in fact, they are equal to 0.0002, 0.0114, and 0.0021, respectively) for P/T (unlike P/PobT, with *p* values of 0.0585, 0.1602, and 0.5909, respectively). Meanwhile, the *p* value for the aforementioned pollen type is greater than 0.05 (specifically equal to 0.6681) for P/PobT, as is the case for P/T (with a *p* value of 0.6169).

However, even though the difference between the coefficients of determination found is only 0.5451% higher between P/T and P/PobT, for *Platanus*, the best fits are obtained with P/T. At the same time, the *p* value for O_3_ is less than 0.05 (specifically 0.0054) in the case of P/T for *Platanus,* a circumstance that does not occur with P/PobT (0.2813). As for the rest of the variables, there is no advantage with respect to their individual contribution to the model.

Table 5 shows the results of the assessment of the normality of the residuals, i.e., the differences between the observed values (observations) and the values predicted by the calculated models. To do this, a normal distribution is fitted to the residual data found previously.

In addition, Table 5 shows the results of the analysis of variance of the residuals, considering the five years of the study as factors.

As can be seen in Table 5, all models calculated for P/T show a normal distribution of residuals, since the *p* values in the goodness-of-fit tests performed are greater than 0.05, with a confidence level of 95%. The same occurs for P/PobT with the pollen type *Platanus* (with a *p* value of 0.1088), contrary to what occurs with Cupressaceae and *Olea* with the same dependent variable P/PobT (with *p* values of 0.0261 and 0.0228, respectively).

At the same time, the *p*-value of the F-ratio in the six calculations performed is greater than 0.05 (in fact, it is equal to 1 in all cases), so there are no statistically significant differences in the mean values of the residuals between the different years of the 2017–2021 period.

Furthermore, given that the *p*-value in Levene’s variance verification test performed for the two types of response variables, P/T and P/PobT, and with the three pollen types studied, is greater than 0.05 in all calculations performed, it can be stated that there is no statistically significant difference between the standard deviations of the residuals within the different years of study, with a confidence level of 95%.

## 5. Discussion

In recent decades, there has been a global trend towards an increase in the prevalence of allergic diseases, such as rhinoconjunctivitis and asthma. At the same time, pollen is the major agent of allergic sensitisation and the main source of airborne allergens globally, particularly in the CAM [1,2,3].

Tree species emit the largest fraction of pollen released into the atmosphere by wind-pollinated plant species in the CAM (73.5%), and in recent decades there has been a clear trend towards an increase in the abundance of tree pollen, among which the most prominent are the pollen types in this study, *Platanus*, Cupressaceae and *Olea* [2,8].

The highest average annual percentage of atmospheric pollen between 1979 and 1993 in Madrid was for *Quercus* (17%), followed by *Platanus* (15%), Gramineae (15%), Cupressaceae (11%) and *Olea* (9%), among others, according to the study conducted by Subiza, J. et al. in 1995 [33]. The most predominant pollens between January and April in the Madrid atmosphere are those from trees, including Cupressaceae and *Platanus*, as well as *Alnus*, *Fraxinus*, *Ulmus*, *Populus* and *Morus* [33].

Between May and June (the annual period with the highest concentration of allergenic pollen in the atmosphere), the most prevalent allergenic pollen was that of *Olea*, followed by *Quercus* and *Pinus*. Meanwhile, the allergenic pollen with the highest prevalence of positive skin test results in Madrid between 1979 and 1993 was Gramineae (94%), followed by *Olea* (61%), *Plantago lagopus* (53%), *Platanus* (52%) and *Cupressus arizonica* and/or *C. sempervirens* (23%). Therefore, during that period, and as has been subsequently confirmed in the CAM, as mentioned above, the most abundant pollen in Madrid is that of tree species [2,8], and the main cause of allergic sensitisation, after that emitted into the atmosphere by grasses [33].

Although the positive results in the skin test for *Olea*, *Platanus* and *Cupressus arizonica* and/or *C. sempervirens* were as indicated (61%, 52% and 23%, respectively), allergic sensitisation induced in isolation by these types of pollen is rare [11,33]. In fact, in the case of *Olea*, which has been described by various authors as a major cause of pollinosis in the Mediterranean area, in almost all cases reflected in the 61% of positive skin tests, there is also a positive reaction to grass pollen, so monosensitisation to *Olea* was very rare, which is why it is very difficult to establish its clinical relevance in the Madrid area [33].

As for *Platanus* pollen, throughout the period 1979–1993, in the study conducted by Subiza et al., a high percentage of allergic patients tested positive for it, 52% as noted. At the same time, it may appear that the allergenic potency of this type of pollen is not very high, since only the most atopic patients, and therefore those who are polysensitised, test positive in skin tests, as shown in the analysis carried out by Nuñez-Borque, E. et al. [6].

Furthermore, in the study conducted by Varela, S. et al., which concludes that *Platanus* pollen is a major cause of pollinosis in Madrid, only 10.9% of patients sensitised to *Platanus* pollen are exclusively sensitised to it; the rest are polysensitised [34]. This polysensitisation is also cited in the study by Subiza, J et al. However, the high percentage obtained in the study by Subiza, J. et al., together with the importance of the symptoms that *Platanus* pollen can cause, mainly allergic rhinitis and asthma, at least in the Madrid area, may suggest that vigilance is needed so as not to underestimate its importance [33].

In accordance with the above, it can be hypothesised that sensitisation to *Platanus* pollen increases the likelihood of allergy to other types of pollen and could be considered a marker of polysensitisation, although this hypothesis needs to be confirmed by studies conducted outside the Madrid area [6].

Cupressaceae airborne pollen has been recognised as a cause of pollinosis, both in the Mediterranean area and beyond. The amount of this pollen recorded is very significant and seems to be experiencing an upward trend driven by plantations mainly of *C. arizonica*, especially in the northern area of Madrid in recent years. Although the response of allergic patients to this type of pollen was positive, with a frequency of 23%, it was also positive to grass pollen at the same time, in a high percentage [33].

The novelty of this research lies mainly in the fact that, for the first time in the Madrid region, results have been obtained that lead to the conclusion that there is a significant interrelationship between the concentrations of pollen types and the atmospheric pollutants studied (independent variables) and allergic sensitisation to these pollen types (dependent variable), and also in the different areas and years studied.

Specifically, the results obtained in this study, in the CAM, in the period 2017–2021, reveal a statistically significant joint influence of pollen concentrations of the Cupressaceae, *Olea* and *Platanus* types, and of the atmospheric pollutants PM_10_, NO_2_ and O_3_, on allergic sensitisation to the respective pollen types. This influence occurs with high percentages of explained variability in five of the six models calculated. Specifically, in the three calculated for the dependent variable P/PobT, for each of the pollen types studied. And in P/T only for Cupressaceae and *Platanus*. The exception is in P/T for *Olea*, as the adjusted R^2^ value is significantly lower than the rest of the adjustments (54.5993%), although it can be classified as an average value. However, in P/PobT for *Olea*, the result of the calculated model adjustment is very high (92.1093%) and, therefore, very similar to the rest of the adjustments obtained for Cupressaceae and *Platanus*.

The reason for this discrepancy is believed to lie in the fact that the number of specimens of *Olea europaea* is significantly lower than that of the other two tree taxa in this study, i.e., the number of trees of the Cupressaceae species and *Platanus hispanica*, a fact that is true in all five study areas. This could influence the pollen load of *Olea,* and, in turn, the results obtained in the skin sensitisation and IgE tests, obtained through blood analysis. It is understood that this cannot be the reason because then we would also expect a significantly lower coefficient of determination for P/PobT than that obtained in this study.

On the other hand, the number of *Platanus hispanica* trees exceeds that of the members of the Cupressaceae family in all areas except Getafe. The highest proportion of *Olea* and *Platanus* trees in relation to the total number of specimens is found in Aranjuez, followed by Getafe and Leganés, respectively. The latter municipality has the highest tree density within the exclusively urban area of the five study areas and has the largest number of trees inventoried in its territory.

It is becoming increasingly necessary to study aerobiological allergenic agents in conjunction with the factors that alter and enhance their allergenicity, i.e., atmospheric pollutants, as well as other factors such as meteorological parameters. The gaseous components (such as NO_2_ and O_3_) and particulate matter (such as PM_10_ particles) analysed in this study, which are part of environmental pollution, especially in cities, lead to an increase in the expression of allergenic proteins in various types of pollen [14]. This fact may help to point to the objective of establishing, from an epidemiological perspective, a measurement system capable of quantifying the increase in the allergenicity of pollen altered by atmospheric pollutants, i.e., one that assesses the degree of risk to citizens’ health [14].

The concentration of pollen grains in the atmosphere can be used as an indicator of the health of plant masses, since, although pollen grains have resistant exines, atmospheric pollutants are the most harmful factors for their viability and survival [8], as these can cause the outer wall of pollen grains to break down, releasing allergenic sub-particles that can even enter the human respiratory tract, especially when they are very small [17,18].

These physical effects can be combined with the chemical effects of protein nitration and oxidation, as well as lipid and nucleic acid oxidation, with the consequent alteration of germination and modulation of the immune response in many cellular and molecular processes. In addition to the formation of adjuvant components in the allergic response, with pro-inflammatory properties. And finally, the biological effects, which consist of modifying the germination rate and viability, two key aspects of plant reproductive function [17,18].

Some studies conducted with asthmatic patients have revealed that O_3_, NO_2_, and particles emitted by internal combustion engines, including PM_10_ (the three pollutants included in this study), cause an increase in the permeability of the bronchial epithelium, thereby facilitating access to the submucosa of inhaled irritants and aeroallergens, where they interact with fibroblasts and smooth muscle cells in the airways. These pollutants also cause a decrease in ciliary beat rate, with a consequent delay in the elimination of the aforementioned agents and exacerbation of the patient’s symptoms [14].

This study shows the existence of an interrelationship between sensitisation to Cupressaceae, *Olea* and *Platanus* pollen and three major atmospheric pollutants, PM_10_, NO_2_ and O_3_. This is true in all five areas and in the different years of the study. Given this interaction between the different variables involved in this research, it is interesting to look at the possible causes that may lead to an increase in the prevalence of allergic sensitisation to the three types of pollen in the CAM population.

Tree pollen types are among the leading causes of allergic sensitisation, both in patients with extrinsic asthma and in those suffering from allergic rhinoconjunctivitis [1,2,4]. And, normally, in the CAM, there is an epidemic spike in asthma episodes in May and June in patients allergic to pollen, although mainly caused by Gramineae, but also produced by tree pollen, including *Olea*, Cupressaceae and *Platanus* [4,35]. On the other hand, the percentages of sensitisation to pollen types, in general, have increased, as have those for *Olea*, Cupressaceae and *Platanus* in the CAM, as can be seen by comparing two studies carried out by the SEAIC in 2005 and 2015 [3].

In asthmatic patients, based on various in vitro studies conducted on bronchial epithelial cells, it can be stated precisely that the air pollutants under study, i.e., O_3_, NO_2_ and the particles released by diesel engines, including PM_10_, act as modulators of different inflammatory mediators at the level of the aforementioned cells [14].

Furthermore, thanks to research carried out on human B cells, it has been proven that the aforementioned particles, together with the polyaromatic hydrocarbons derived from them, can stimulate the synthesis of IgE, together with the cytokine interleukin 4 (IL-4), resulting in stimulation of sensitivity to aeroallergens, thereby producing a direct adjuvant effect on the part of these pollutants [14].

Both cytokines and chemokines released as a result of the action of these pollutants can ultimately lead to tissue damage, increased airway permeability, and therefore enhanced penetration of pollen aeroallergens into the submucosa, where they can interact with eosinophils, lymphocytes, neutrophils, recruited inflammatory mast cells, as well as innate lymphoid cells and resident dendritic cells, resulting in the manifestation of symptoms characteristic of an allergic reaction [5,14].

In the three types of pollen analysed, there is a high dispersion of concentration values around their mean values, which seems to point to the fact that pollination is mainly concentrated in more or less defined periods of time (the main pollination periods), which in the case of *Platanus* is limited to a period of 2 to 4 weeks and normally begins in the second half of March. During the rest of the year, its airborne concentration values decrease significantly. Furthermore, this type of pollen is very striking in terms of its production, which exceeds that of all other types of pollen present in the atmosphere of the CAM [8,19].

The frequency of sensitisation to *Platanus* allergens may vary depending on the geographical area in question, due to atmospheric pollutants, as well as meteorological factors such as wind, temperature and humidity, and pruning regimes [6].

In contrast to pollen concentration values, the coefficients of variation for the three atmospheric pollutants studied are much lower. It can be interpreted that the greater stability of these values in relation to those of pollen types is since the human activities that generate them (mainly combustion processes) remain more constant throughout the year.

The same conclusion can be drawn from another study conducted in the CAM, which analysed the interrelationships between atmospheric emissions of pollen and atmospheric pollutants in the period 2013–2017 [19]. In this research, the average values and dispersion measurements of O_3_, NO_2_, and PM_10_ are very similar to those obtained in the present study, and the average values for Cupressaceae and *Platanus* are also practically identical (with somewhat similar dispersion measurements), while those for *Olea* are somewhat more divergent between the two studies, albeit with very similar coefficient of variation values [19].

Specifically, that study found that O_3_ and NO_2_, followed by PM_10_, are the pollutants most strongly correlated with the pollen types mentioned. And that the highest positive correlations in the 2013–2017 period, in the CAM, occur with O_3_ and pollen from *Olea* and *Platanus*, as well as with *Pinus*. The highest negative correlations were found between NO_2_ and pollen from *Olea* and *Pinus*. *Olea*, Cupressaceae and *Platanus*—precisely the most allergenic in that study and in the present one—are among the pollen types with the highest number of significant correlations with the atmospheric pollutants O_3_, NO_2_ and PM_10_, as well as CO, in the different areas of that study, which coincide with those of the Palinocam Network, including those included in the present investigation [19].

A study conducted on the effects of air pollution on pollen aeroallergens shows that exposure of *Platanus* pollen to gaseous pollutants can lead to an increase in the levels of Pla a 1, its main allergen, because of the increase in total pollen protein content [36]. It also shows that atmospheric pollutants can cause weakening and fragility of the exine in Cupressaceae pollen, with the consequent potentiation of its allergenicity and an increase in the expression of the allergen Cup a 3. This is the second main allergen of *Cupressus arizonica* (a species considered to be the source of the main concentrations of Cupressaceae pollen in the atmosphere of the CAM [8]), present in the walls and cytoplasm of pollen grains during the stages of hydration and atmospheric dispersion [12,36,37,38].

Research carried out in different Spanish regions confirms that, under conditions of atmospheric pollution, specimens of *Cupressus arizonica* intensify their activity as a defence against atmospheric pollutants, which implies an exacerbation of their allergenicity [12]. This confirms the conclusions of several recent studies, which state that polluted air acts as a stressor that increases the expression of certain allergens contained in pollen grains [36].

Sensitisation to Cupressaceae pollen, and, in general, to coniferous tree pollen of the order Pinales, has increased substantially in recent years, although the prevalence of sensitisation has not yet been studied in depth in Spain. Furthermore, the CAM can be defined as an area of high exposure to *Cupressus arizonica* pollen [11]. Allergy to Cupressaceae pollen in Madrid was already confirmed in a study conducted on patients with symptoms of pollinosis in the winter season and with a positive skin test for *Cupressus arizonica*, in which total IgE was measured and specific IgE for *Cupressus arizonica* and *Cupressus sempervirens* was determined [39].

The abundant planting of cypress trees, particularly in the CAM region, but also widespread throughout the Mediterranean basin in recent decades, combined with environmental pollution, which interacts with pollen grains, has led to an increase in the allergenicity of Cupressaceae pollen [7]. Likewise, the in vivo allergenicity, assessed by skin testing, of pollen extracts from *Cupressus arizonica* is higher in polluted areas than in areas without air pollution [36].

Allergic sensitisation to Cupressaceae pollen is highly prevalent among the causes of allergic rhinitis [40], in addition to other pathologies such as asthma [3]. The study, which, like this one, was conducted on patients from five hospitals near Madrid shows that the symptoms experienced by patients sensitised to conifer pollen are often those of rhinoconjunctivitis, although more recently, symptoms of asthma have also been detected. In addition, 88% of patients sensitised to this pollen are also sensitised to other types of pollen [11]. In this study, all patients with positive sIgE results for *C. arizonica* pollen also had a positive sIgE level for its main allergen, Cup a 1 [7,11]. And the IgE values for Cup a 1 were significantly higher than for the complete extract of *C. arizonica* [11].

It is interesting to note that, as demonstrated by the study conducted on *Cupressus arizonica* pollen, flow cytometry allows both complete pollen count and allergen load to be measured, as an alternative to measurement using optical microscopy [40].

This study therefore implies that, in addition to the complete pollen population of *Cupressus arizonica*, there is a population of smaller, incomplete particles [40]. And that both populations (which together make up the pollen aeroallergens of *C. arizonica*) can induce allergic sensitisation to this species of cypress [40]. This is because, since these pollen particles from *C. arizonica* are recognised by the polyclonal rabbit antibody Cup a 1, using flow cytometry, these particles may be responsible for the symptoms experienced by patients allergic to *C. arizonica* pollen [40].

According to research conducted on cypress and olive pollen, double allergic sensitisation to Cupressaceae and *Olea* pollen types, without sensitisation to any other type of pollen, is relatively common in the CAM (2% of all allergic patients in that area suffer from it) and persists over time [7]. This is due to co-sensitisation to the allergens Cup s 1 and Ole e 1, and to cross-reactivity between Ole e 1, Ole e 9 and Ole e 11 and homologous allergens in cypress pollen, not previously described [7]. This double sensitisation results in four clinical phenotypes of seasonal/perennial respiratory allergy, i.e., rhinoconjunctivitis with and without asthma, occurring in winter and/or spring or year-round [7].

In the study conducted in the Cova da Beira region of Portugal, significant differences in allergic sensitisation to herbaceous plant pollen (Gramineae, in addition to *Parietaria judaica*) and *Olea europaea* were observed within different age subgroups of the allergic population when comparing urban and rural exposures. Specifically, the frequency of sensitisation to olive tree pollen aeroallergens, when comparing the urban and rural environments, was 30.2% and 23.3%, respectively. This study also shows that sensitisation to indoor airborne allergens is lower than to pollen and furthermore concludes that atmospheric pollutants can increase sensitisation in urban areas [41].

It is also interesting to analyse how the pollutants in this study, such as NO_2_, may influence other pollen aeroallergens not included in the study, which can be classified as ‘panallergens’. In Spain, fruits and nuts, followed by legumes and fresh vegetables, are the foods that cause the highest frequency of food allergies in people over 5 years of age, among whom more than 75% are allergic to pollen, with variations in pollen type depending on the aerobiology of the area [42].

This cross-reactivity is determined by the presence of IgE antibodies against ‘panallergens’, proteins that are widespread in the plant kingdom and have biologically important functions, normally related to defence. One of the three best-known groups of these panallergens—along with profilins and lipid transfer proteins (LTPs)—is made up of homologous allergens of Bet v 1, the major allergen in birch pollen [42].

The study conducted on the nitration of the main protein in *Betula* pollen shows that NO_2_ is an agent that enhances the allergenicity of Bet v 1, as well as its affinity for IgE through the formation of nitrotyrosine residues, a product of tyrosine nitration mediated by reactive N species [36,43]. In this study, conducted with serum from mice sensitised to the Bet v 1 allergen, with and without nitration, it was found that the nitrated samples showed higher levels of specific IgE than the non-nitrated samples. In addition, higher levels of functional specific IgE were found against nitrated Bet v 1 than against the same untreated allergen in the sera of patients allergic to birch pollen, which is why nitration caused by NO_2_ in cities such as those included in this study may lead to a higher prevalence of allergic diseases [43].

On the other hand, a research conducted in Mexico City shows that high increases in pollen counts of *Fraxinus* and Cupressaceae correlate with increases in the frequency of sensitisation to these pollen types in patients with allergic rhinitis and allergic rhinitis associated with asthma. Furthermore, this fact correlates with increases in the concentration of two of the pollutants in this study, namely PM_10_ and NO_2_ [44].

In addition, NO_2_ and O_3_ can damage the cell membranes of pollen in the sub-pollen particles released from them, thereby increasing the amount of the allergen Pla a 3 emitted into the atmosphere, as demonstrated by the experiment conducted by a study on *Platanus* pollen [45]. Likewise, these atmospheric pollutants could alter, through oxidation and nitrification, the structure of the Pla a 3 protein, which leads to an increase in the immunogenicity and stability of these aeroallergens. In this experiment, a significant worsening of pollen-induced allergic pneumonia in animals was also demonstrated through an in vivo trial [45].

Although it can be said that this research shows a significant correlation between the pollen types and atmospheric pollutants studied, with allergic sensitivity to these types of pollen in the different years and study areas, as has been pointed out, there are also limitations that must be considered.

The first limitation is the lack of knowledge about the clinical condition of the patients participating in the study, who remain anonymous. Although the results of the studies carried out by the SEAIC [3] are known, it would be very interesting to know the pathologies that affect them to reach more accurate conclusions.

It is also worth mentioning meteorological factors whose effect on pollen, pollutants and the patients themselves have not been evaluated in this study.

On the other hand, only strictly urban areas of the CAM have been considered in this research, albeit with different population densities. It could be very interesting to conduct a study that also includes clearly rural geographical areas, to establish a contrast with the results obtained in this study.

Finally, it is worth noting that this is a retrospective study, covering a specific period. This provides very valid and solid results, which bring us closer to the reality of pollen allergy awareness in the CAM. However, it does not take advantage of the benefits of a prospective study, such as being able to establish a clearer cause-and-effect relationship or having greater control over the research process.

## 6. Conclusions

Based on the results obtained, as well as the contrast that can be established between them and the results of other research carried out on the subject matter addressed in this study, as reflected in the previous section, it can be stated that:-The pollen types studied are among the most prominent agents of allergic sensitisation in the CAM, with a clear upward trend in sensitisation to the most prevalent extrinsic pathologies, namely rhinoconjunctivitis and asthma.-The prevalence of sensitisation to Cupressaceae, *Olea* and *Platanus* pollen types in the CAM region in the period 2017–2021 is statistically significantly correlated with the concentrations of these pollen types and of the atmospheric pollutants PM_10_, NO_2_ and O_3_, as well as with the different years and study areas. Both pollen and atmospheric pollutant concentrations can be influenced by geographical location and meteorological parameters.-The independent variables jointly influence the prevalence of sensitisation, with medium-high percentages of variability explained by all the models calculated.-Although all the models calculated can explain the variability of pollen sensitisation (dependent variable) in a very significant way, in the case of P/T with *Olea*, the adjusted R^2^ value is significantly lower than that of the other adjustments. It is true that the number of specimens of *Olea europaea* is significantly lower than that of the other two types of pollen in the five study areas. However, the adjustment made for the P/PobT variable for *Olea* yields a significantly higher result, similar to that of the other models obtained in the study.-Although it was not possible to obtain clinical information from the study patients regarding pollen-induced rhinitis, conjunctivitis and asthma, it would be very useful to have access to this information in the future in order to complete the data on allergic sensitisation to Cupressaceae, *Olea* and *Platanus* pollen provided by the five hospitals that participated in this research.-The municipality of Leganés has the highest tree density in relation to its urban area, followed by Aranjuez. Leganés also has the highest number of trees in its territory, followed by Getafe.-The highest proportion of *Olea* and *Platanus* trees in relation to the total number of specimens is found in Aranjuez, followed by Getafe and Leganés, respectively. Alcobendas has the highest proportion of Cupressaceae trees, ahead of Getafe. In all study areas except Getafe, *Platanus* is the taxon with the highest number of trees.-The highest peaks in *Olea* pollen concentration occur, in descending order, in Getafe, Aranjuez, Leganés, Alcobendas and Madrid’s Salamanca district, coinciding with and corresponding to the number of *Olea europaea* trees inventoried in the five areas.-At the five study stations, the peak values of the P/T and P/PobT sensitisation curves (less noticeably) for *Olea* occur in the months immediately following the pollination periods.-It is considered worthwhile to conduct research on allergic sensitisation in the CAM region that includes meteorological factors, in addition to those involved in this study.-Furthermore, future research could delve deeper into a study based on allergic sensitisation data combined with clinical prevalence data from the same patients.

## Figures and Tables

**Figure 1 life-15-01774-f001:**
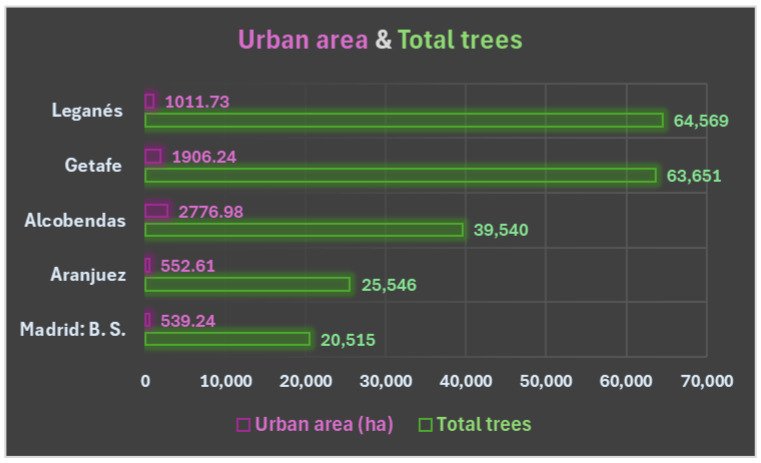
Specification of the total number of trees inventoried in the five study areas and the urban area in which these trees are located.

**Figure 2 life-15-01774-f002:**
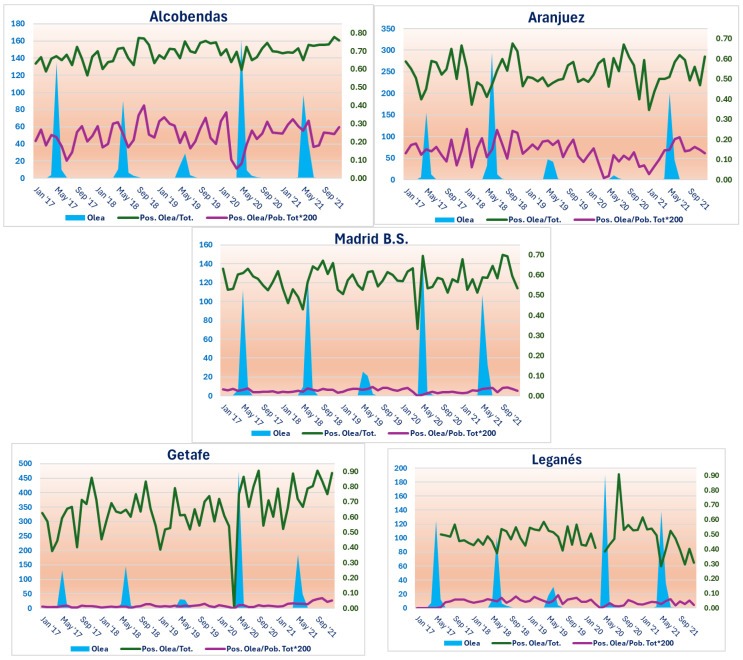
Relationship between the five graphs representing *Olea* pollen sensitisation and the expression of the dependent variables P/T (green line) and P/PobT (purple line) in relation to the concentrations of this pollen type in the atmosphere of the CAM (sky blue areas) during the period 2017–2021, in the five study areas. The values of the P/PobT variable have been multiplied by 200 in order to appreciate its evolution over time. The vertical axis on the left expresses the pollen concentration in g/m^3^, and the one on the right, the P/T and P/PobT ratios (×200).

**Table 1 life-15-01774-t001:** Names of the stations in the Palinocam Network, the CAM Hospital Centres (from which the data on sensitivity to the types of pollen studied originate), and the Air Quality stations. The approximate distances in metres from each hospital and each air quality station to the corresponding Palinocam Network station, measured using the Google Earth geographic information system, are shown in brackets. The area corresponding to the capital city of Madrid is shaded in light blue.

Palinocam Network Stations	Hospital Centres	Air Quality Network Stations
Alcobendas	Infanta Sofía University Hospital (3970 m)	Alcobendas (693 m)
Aranjuez	Tajo University Hospital (2930 m)	Aranjuez (634 m)
Madrid: Barrio de Salamanca	Gregorio Marañón University General Hospital (2000 m)	Escuelas Aguirre (1718 m)
Getafe	Getafe University Hospital (800 m)	Getafe (1436 m)
Leganés	Severo Ochoa University Hospital (930 m)	Leganés (1586 m)

**Table 2 life-15-01774-t002:** Names of the CAM Health Centres, which serve the total assigned population in each of the five study areas (shown on the right-hand side of the table, by year), defined by the areas of influence of the five stations in the Palinocam Network covered by this research. The approximate distances of each Health Centre from the corresponding Palinocam Network station, measured using the Google Earth geographic information system, are shown in brackets. The only area corresponding to the capital of Madrid is shaded in light blue.

Palinocam Network Stations	Nearest Health Centres	Total Assigned Population by Year of Study				
		2017	2018	2019	2020	2021
Alcobendas	Marqués de Valdavia (630 m), Valdelasfuentes (880 m), La Chopera (1130 m), Miraflores (1420 m), Arroyo de la Vega (1890 m).	116,918	118,289	119,925	118,665	120,335
Aranjuez	Aranjuez (0 m), Las Olivas (1400 m).	60,920	61,873	62,490	61,638	61,590
Madrid: Barrio de Salamanca	Príncipe de Vergara (515 m), Lagasca (585 m), Montesa (815 m), Castelló (939 m), Prosperidad (1070 m), Londres (1118 m), Santa Hortensia (1150 m), Ciudad Jardín (1332 m), Baviera (1392 m), Espronceda (1685 m), Goya (1734 m), Segre (1761 m), Daroca (1770 m), Ibiza (2003 m), Eloy Gonzalo (2044 m), Justicia (2110 m), Canal de Panamá (2170 m), Potosí (2290 m).	480,329	481,580	483,804	474,001	478,636
Getafe	Las Margaritas (784 m), Juan de la Cierva (816 m), Sánchez Morate (940 m), El Greco (1332 m), Las Ciudades (1693 m), El Bercial (1783 m), Getafe Norte (1969 m), Sector III (2050 m), Perales del Río (6584 m).	184,680	186,065	188,445	186,385	186,857
Leganés	Santa Isabel (153 m), Doctor Mendiguchia Cariche (765 m), María Jesús Hereza (920 m), Huerta de los Frailes (1195 m), María Ángeles López Gómez (1460 m), Jaime Vera Leganés (1705 m), María Montessori (2070 m), Leganés Norte (2375 m), Marie Curie (3690 m).	193,291	194,151	194,965	191,392	190,609

**Table 3 life-15-01774-t003:** Mean values, standard deviation (in pollen grains per cubic metre of air, g/m^3^), and coefficient of variation (in percentage), during the study period (2017–2021), for the three pollen types and three atmospheric pollutants studied at the five stations of the Palinocam Network already reflected in Table 1 and described above.

**Pollen Types** **(Size Major Axis in μm)**	**Mean**	**Standard** **Deviation**	**Coefficient of Variation (%)**
Cupressaceae (22–28)	32.2254	65.5155	203.304
*Olea* (18–25)	13.6864	46.4434	339.339
*Platanus* (17–22)	47.1563	159.515	338.269
**Airborne** **Pollutants (µg/m^3^)**	**Mean**	**Standard** **Deviation**	**Coefficient of Variation (%)**
Ozone (O_3_)	52.0809	20.2463	38.8748
Nitrogen dioxide (NO_2_)	31.4869	16.0731	51.0470
Particles < 10 µm (PM_10_)	19.7984	6.3897	32.2739

**Table 4 life-15-01774-t004:** Number of observations and results of the *p* values of the general linear statistical models calculated for the two types of dependent variables (P/T and PobT) as well as the factors involved in them, and the values of the adjusted coefficient of determination R2, obtained for the entire period 2017–2021 and at the five study stations.

	Cupressaceaae		*Olea*		*Platanus*	
	P/T	P/PobT	P/T	P/PobT	P/T	P/PobT
**No. of observations:**	283	269	282	269	285	279
** *p* ** **-value model:**	0.0000	0.0000	0.0000	0.0000	0.0000	0.0000
** *p* ** **-value categorical factors:**						
Station:	0.0000	0.0000	0.0000	0.0000	0.0000	0.0000
Year:	0.0000	0.0004	0.2854	0.0000	0.0001	0.0010
** *p* ** **-value pollen types:**	0.6169	0.6681	0.1152	0.4942	0.1019	0.9200
** *p* ** **-value atmosph. pollutants:**						
O_3_:	0.0002	0.0585	0.7948	0.5064	0.0054	0.2813
NO_2_:	0.0114	0.1602	0.7948	0.3462	0.3537	0.3100
PM_10_:	0.0021	0.5909	0.8398	0.3151	0.0634	0.7672
**Adjusted R^2^:**	84.0234%	92.4880%	54.5993%	92.1093%	81.8703%	81.3252%

**Table 5 life-15-01774-t005:** Results of the normality and variance tests, with *p* values relating to the goodness-of-fit test, as well as the F ratio and variance verification, obtained for the entire period 2017–2021.

*p* Values	Cupressaceaae		*Olea*		*Platanus*	
	P/T	P/PobT	P/T	P/PobT	P/T	P/PobT
**Kolmogorov–Smirnoff:**	0.9482	0.0261	0.5889	0.0228	0.7633	0.1088
**F-ratio:**	1.0000	1.0000	1.0000	1.0000	1.0000	1.0000
**Variance verification** **(Levene’s):**	0.0891	0.8799	0.0616	0.4872	0.9211	0.9817

## Data Availability

All data used in this study, except those relating to patients, are included in the article. The latter are not available due to ethical or privacy restrictions.

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
