# Peer review of "Influence of Atmospheric Pollutants on Allergic Sensitization to Cupressaceae, *Olea*, and *Platanus* Pollen in the Community of Madrid (2017–2021)"

_life, 2025, doi:10.3390/life15111774_

Round 1
Reviewer 1 Report
Comments and Suggestions for Authors
In this paper, the Authors explore the relationship between air pollutants, major tree pollens in the CAM region, and allergic diseases. The topic, in itself quite relevant, is explored via a multiple-source data collection, offering a faceted exploration of the environmental determinants of allergic sensitization.
Unfortunately, while conceptually sound and data-rich, the paper is marred by a number of points of concern undermining its clarity, focus and data interpretation.
General consideration: as a whole, the paper suffers from readability issues. The text appears as a mixture between a review and a technical report, especially the introduction and "materials and methods" sections. Sentences are overly long, with heavy use of punctuation and concatenation. Furthermore, data is often presented as large "information dumps" that, albeit commendably detailed, subtract from the reader's attention and stray from the paper's main scope. Please consider a major reworking of the paper's flow, ideally by a native English speaker: aim for conciseness and a hypothesis driven structure, focusing on biological rationale, key models output and trends, and clinical repercussions of your findings within the available literature corpus.
Lines 189-191: the paper's aim is, simply put, far too concisely described - especially when compared to the extremely detailed discussion of the CAM region's pollens and the role of Particulate Matter in mediating tissue damage, sensitization and allergic reactions. Please consider reworking the passage to clarify the Authors' intent.
Line 220: please provide the protocol numbers for both Committees' ethical approval.
Lines 241-269: this section and its contents appear more suited for discussing the study's results, or as part of the introduction (e.g. demonstrating which pollens are prevalent in which allergic disease) - consider revising its placement. Please note that, as previously stated, there is a remarkable lack of conciseness and heavy usage of convoluted sentences. The section is far too densely packed with a plethora of information that is very often not strictly needed for the paper's matter at hand. Consider moving a considerable portion to Supplementary Materials.
Lines 417-423: this section is especially convoluted, with repeated information. Please revise.
Discussion: as a whole, the section shows instances of descriptive repetition. Discuss limitations more explicitly: lack of control for confounders (e.g., meteorology, urban density), retrospective design, and lack of molecular validation.
Comments on the Quality of English LanguagePlease see the above section.
Author Response
In this paper, the Authors explore the relationship between air pollutants, major tree pollens in the CAM region, and allergic diseases. The topic, in itself quite relevant, is explored via a multiple-source data collection, offering a faceted exploration of the environmental determinants of allergic sensitization.
Unfortunately, while conceptually sound and data-rich, the paper is marred by a number of points of concern undermining its clarity, focus and data interpretation.
General consideration: as a whole, the paper suffers from readability issues. The text appears as a mixture between a review and a technical report, especially the introduction and "materials and methods" sections. Sentences are overly long, with heavy use of punctuation and concatenation. Furthermore, data is often presented as large "information dumps" that, albeit commendably detailed, subtract from the reader's attention and stray from the paper's main scope. Please consider a major reworking of the paper's flow, ideally by a native English speaker: aim for conciseness and a hypothesis driven structure, focusing on biological rationale, key models output and trends, and clinical repercussions of your findings within the available literature corpus.
Thank you for your comments and suggestions, which contribute to the improvement of this study. Overly long sentences have been divided and many expressions have been replaced with more concise ones throughout the manuscript in order to improve the readability and comprehension of this study. Some paragraphs have been rewritten for this purpose, as you will see.
- Lines 189-191: the paper's aim is, simply put, far too concisely described - especially when compared to the extremely detailed discussion of the CAM region's pollens and the role of Particulate Matter in mediating tissue damage, sensitization and allergic reactions. Please consider reworking the passage to clarify the Authors' intent.
Thank you for your suggestion. Information about the purpose and motivations behind the article has been added (lines 239–242). This purpose is complemented in the Discussion section (lines 797–801).
- Line 220: please provide the protocol numbers for both Committees' ethical approval.
Agreed. The protocol codes for both ethics committees are provided, as per your suggestion (lines 279–280).
- Lines 241-269: this section and its contents appear more suited for discussing the study's results, or as part of the introduction (e.g. demonstrating which pollens are prevalent in which allergic disease) - consider revising its placement. Please note that, as previously stated, there is a remarkable lack of conciseness and heavy usage of convoluted sentences. The section is far too densely packed with a plethora of information that is very often not strictly needed for the paper's matter at hand. Consider moving a considerable portion to Supplementary Materials.
Thank you for your comment. In fact, this section serves as the basis for the Discussion section, as is the case with the description of tree data in the following section. It is considered highly relevant information, specific to the CAM, from a highly accredited source, the SEAIC.
Furthermore, as mentioned above, the manuscript has been reviewed and corrected in order to identify overly long sentences that make it difficult to understand the content. We appreciate you pointing this out, as in your next suggestion.
- Lines 417-423: this section is especially convoluted, with repeated information. Please revise.
Thank you. It has been rewritten to make it easier to understand (lines 485-490).
- Discussion: as a whole, the section shows instances of descriptive repetition. Discuss limitations more explicitly: lack of control for confounders (e.g., meteorology, urban density), retrospective design, and lack of molecular validation.
Thank you again for your comment and suggestion. We did not observe any descriptive repetition in the Discussion section. It merely makes explicit information corresponding to the preceding sections, which serve as the basis for developing the Discussion of the research. We would be grateful if you could indicate exactly which repetitions you are referring to.
Furthermore, some paragraphs has been added at the end of this section, outlining the limitations of the study, in accordance with your suggestion (lines 1020-1038).

Reviewer 2 Report
Comments and Suggestions for Authors
The manuscript investigates the combined influence of three atmospheric pollutants and three types of tree pollen on the prevalence of allergic sensitization in five areas of the Community of Madrid. The authors employ general linear models using a five-year dataset and report statistically significant models that explain a medium-to-high percentage of the variability in sensitization. They conclude that sensitization is significantly correlated with pollen and pollutant concentrations, which vary by year and geographical area. The review is well written and structured.
However, there are a number of comments:
- The abstract should be revised to include the key findings more directly.
- The manuscript does not sufficiently articulate how it advances beyond existing research. The authors must explicitly state the novel aspect of their work in the introduction and discussion. Is it the specific 2017-2021 timeframe, the particular statistical modelling of combined effects, or the use of two different sensitization variables? This needs to be clarified to establish the manuscript's significance.
- The introduction provides a general background but would be strengthened by incorporating a more detailed, mechanistic discussion of how pollutants are thought to enhance sensitization, referencing current understanding.
- The manuscript requires thorough proofreading for English grammar and sentence structure to improve readability and ensure it meets the journal's language standards.
- Please check the reported references also in accordance with the format required by “Life-MDPI”.
The English could be improved to more clearly express the research.
Author Response
The manuscript investigates the combined influence of three atmospheric pollutants and three types of tree pollen on the prevalence of allergic sensitization in five areas of the Community of Madrid. The authors employ general linear models using a five-year dataset and report statistically significant models that explain a medium-to-high percentage of the variability in sensitization. They conclude that sensitization is significantly correlated with pollen and pollutant concentrations, which vary by year and geographical area. The review is well written and structured.
However, there are a number of comments:
Thank you very much for your suggestions and comments, which contribute positively to the improvement of the article. Overly long sentences have been divided and many expressions have been replaced with more concise ones throughout the manuscript in order to improve the readability and comprehension of this study. Some paragraphs have been rewritten for this purpose, as you will see.
1.The abstract should be revised to include the key findings more directly.
Thank you for your suggestion. The wording of the key points in the summary has been changed in order to highlight the results obtained in the study (lines 31-35).
2.The manuscript does not sufficiently articulate how it advances beyond existing research. The authors must explicitly state the novel aspect of their work in the introduction and discussion. Is it the specific 2017-2021 timeframe, the particular statistical modelling of combined effects, or the use of two different sensitization variables? This needs to be clarified to establish the manuscript's significance.
Thank you for your comment. The novelty of the study is highlighted, as you suggest, in the Introduction (lines 239–242) and in the Discussion section (lines 797–801).
3.The introduction provides a general background but would be strengthened by incorporating a more detailed, mechanistic discussion of how pollutants are thought to enhance sensitization, referencing current understanding.
Thank you again for your suggestion. The Introduction certainly shows how atmospheric pollutants can influence, more or less indirectly, allergic sensitisation to pollen through various mechanisms.
In addition, in accordance with your suggestion, a more detailed description has been added on important adjuvants such as pollen-associated lipid mediators (PALMs). the action of pollen allergen and adjuvant adhesion to the surfaces of pollutant particles has been included, and the effects of the atmospheric pollutants under study on the processes that lead to increased allergic sensitisation have also been specified (lines 169-170; 172-187; 196-198; 204-205; 210-213 and 234-236).
As can be seen from the various studies referenced in the Introduction, there is mention of its action on vegetation and pollen grains, with the consequent release of allergenic bioparticles (lines 150-154 and 168-171). It also specifies how these agents produce harmful effects on human anatomy and physiology, as well as on the immune system itself (lines 155-160). In addition, their important role as adjuvants is mentioned (lines 163-167).
Furthermore, the action of each of the atmospheric pollutants under study is specified (lines 188 onwards).
In addition, the Discussion section also discusses the mechanisms of allergic sensitisation derived from the action of the pollutants under study (among others, lines 802-812; 849-855).
4.The manuscript requires thorough proofreading for English grammar and sentence structure to improve readability and ensure it meets the journal's language standards.
The manuscript has been reviewed and corrected in order to detect phrases that make it difficult to understand its content. We would appreciate it if you could point this out to us.
5.Please check the reported references also in accordance with the format required by “Life-MDPI”.
Very well, thank you. The references in the manuscript have been reviewed and corrected.

Reviewer 3 Report
Comments and Suggestions for Authors
The paper presents an interesting statistical analysis of the relationship between pollen allergies from 3 tree families and atmospheric pollutants. The main issue is that the paper is too long. There is many unnecessary information, lengthy descriptions and extended discussions on both the statistical analysis, and general aspect of allergy. The authors should keep in mind that their interesting results are based on a simple (though robust) statistical analysis involving only 3 tree species, 5 different locations, and 3 atmospheric parameters. This does not justify nearly 1000 lines of text; a length below 500 would be more appropriate for this type of paper. The reader is quickly overwhelmed by excessive information and repetition, which risks obscuring the nice results. A revised and shortened version of the paper is therefore recommended.
The order of the tables cited in the text does not follow a unit-by unit progression (for example, table 5 cited before table 4).
Lines 76-79: The sentence in unclear and overly complex.
Line 104: This value is surprising. In most cases, analysis concern the number concentrations (number of pollens grains detected per m3). Does “g” refer to grams or grains?
Lines 179-182: Where does this value come from? In most cities, PM10 are manly produced by human activities, so they are not natural. Human activities do not only generate carbon particles from combustion. Construction works, tires, brake wear also produce significant amounts of particles. Please provide reference to international scientific papers.
Line 188: Please provide reference to international scientific papers.
Line 193-199: The sentence is too long, and contains two semicolons.
Table 2: Ano is not an English word
Table 2: How can the authors be confident of such precise tree counts, with accuracy to single tree?
Line 293-309: The authors correctly list the locations not included in their analysis. But what are the consequence of not considering the areas not managed by the local councils?
Part 4: Once decimal place would be sufficient for the numerical results
Table 5: “Stacion” is not an English word.
Lines 597: Can the authors explain the origin of the value “300”?
Line 623 and elsewhere: Why is R2 expressed in percentage?
Lines 620-625: These sentences are unclear and difficult to follow. More explanations is needed.
Part 4.1: This section is far too long, with too many intermediate results. A simplified presentation of the results is recommended.
Part 5: Most of this section repeats information already presented in the introduction (except of the results obtained in the present study). There are too much repetitions and not real discussion supported by new references. The end of this section is more like an introduction, as there is no substantial comparison of the results with previous studies.
Lines 841 and after: Refences in the text must be given in the correct format (number instead of author names).
The conclusion reads more like a summary. Perspectives for future works should be given.
Author Response
The paper presents an interesting statistical analysis of the relationship between pollen allergies from 3 tree families and atmospheric pollutants. The main issue is that the paper is too long. There is many unnecessary information, lengthy descriptions and extended discussions on both the statistical analysis, and general aspect of allergy. The authors should keep in mind that their interesting results are based on a simple (though robust) statistical analysis involving only 3 tree species, 5 different locations, and 3 atmospheric parameters. This does not justify nearly 1000 lines of text; a length below 500 would be more appropriate for this type of paper. The reader is quickly overwhelmed by excessive information and repetition, which risks obscuring the nice results. A revised and shortened version of the paper is therefore recommended.
- The order of the tables cited in the text does not follow a unit-by unit progression (for example, table 5 cited before table 4).
Thank you. References to Table 5 have been removed as they are unnecessary (lines 532 and 539).
- Lines 76-79: The sentence in unclear and overly complex.
Agreed. The wording has been changed so that it can be understood more clearly (lines 84-88).
- Line 104: This value is surprising. In most cases, analysis concern the number concentrations (number of pollens grains detected per m3). Does “g” refer to grams or grains?
Indeed, it refers to pollen grains per cubic metre. The value of 100g/m3 written in the manuscript is therefore correct (line 113).
- Lines 179-182: Where does this value come from? In most cities, PM10 are manly produced by human activities, so they are not natural. Human activities do not only generate carbon particles from combustion. Construction works, tires, brake wear also produce significant amounts of particles. Please provide reference to international scientific papers.
Thank you for your suggestion. This value comes from a study by a group of chemical scientists, with its publication duly accredited and referenced in the manuscript [21]. However, in addition, a new bibliographic reference has been included in response to your request (lines 218-226).
- Line 188: Please provide reference to international scientific papers.
Thank you again. This information comes from a scientific work, as mentioned in the previous response, with reference [21]. However, information from another scientific work has also been added, in accordance with your suggestion (lines 231-233).
- Line 193-199: The sentence is too long, and contains two semicolons.
Thank you. It has been corrected (lines 251-257).
- Table 2: Ano is not an English word
Thank you. It's been corrected.
- Table 2: How can the authors be confident of such precise tree counts, with accuracy to single tree?
Tree counts have been carried out by technical professionals specialising in tree management in the different municipalities included in the study. These technicians have provided the data for this research and continuously update their tree data by updating their inventory databases.
- Line 293-309: The authors correctly list the locations not included in their analysis. But what are the consequence of not considering the areas not managed by the local councils?
This study has followed the criterion of including trees that grow on urban land in the five municipalities included in the study, as explained in the article. In fact, the areas not included are located on non-urban land. These areas are mentioned because of their proximity or adjacency to strictly urban areas. Obviously, criteria must be established for the study area, bearing in mind that the pollen recorded at the measuring stations may come not only from urban areas but also from neighbouring areas and other more distant areas, due to the action of meteorological factors such as wind.
- Part 4: Once decimal place would be sufficient for the numerical results
Thank you. Maintaining four decimal places allows for comparisons of results such as those made in lines 683 and subsequent lines. In order to present the rest of the values with adequate uniformity, they are maintained throughout the Results section.
- Table 5: “Stacion” is not an English word.
Thank you. It's been corrected.
- Lines 597: Can the authors explain the origin of the value “300”?
300 is the maximum number of monthly observations that can occur, taking into account the five stations and the five years of study. This has been clarified in the text (lines 662–663).
- Line 623 and elsewhere: Why is R2 expressed in percentage?
The coefficient of determination can be expressed perfectly as a percentage or as a ratio. Both forms of expression are correct.
- Lines 620-625: These sentences are unclear and difficult to follow. More explanations is needed.
All right, thank you. The content of this paragraph (lines 692-697) has now been explained again.
- Part 4.1: This section is far too long, with too many intermediate results. A simplified presentation of the results is recommended.
Thank you. Section 4.1 has been revised and, as you can see, the results obtained are carefully analysed. Only with a detailed explanation of these results is it possible to understand them properly.
- Part 5: Most of this section repeats information already presented in the introduction (except of the results obtained in the present study). There are too much repetitions and not real discussion supported by new references. The end of this section is more like an introduction, as there is no substantial comparison of the results with previous studies.
Thank you for your comment.
The Discussion section provides information that complements that given in the Introduction. Therefore, it is not the same. Furthermore, the data provided at the beginning of the Discussion section on the types of pollen studied serve to document their relevance in terms of their role in triggering allergic sensitisation in the CAM.
Furthermore, the last paragraphs of the Discussion section, up to line 1019, explain the relationship between the different independent variables (pollen and pollutant concentrations in the atmosphere) in this study and allergic sensitisation, based on various research studies. In fact, for example, the last paragraph describes the interrelationship between two of the pollutants studied and one of the Platanus allergens.
- Lines 841 and after: Refences in the text must be given in the correct format (number instead of author names).
Thank you. It has already been corrected (lines 917 and susequent lines).
- The conclusion reads more like a summary. Perspectives for future works should be given.
Thank you very much. In accordance with your suggestions, two perspectives on future lines of research have been included (lines 1079-1082).

Round 2
Reviewer 1 Report
Comments and Suggestions for Authors
We would like to thank the Authors for the revised version. While we believe the paper to still be overly long and information-dense in some parts (e.g. section 2.2, which delves into overly intricate details that detract from the paper's scope), which would benefit from further streamlining and usage of Supplementary Materials, we believe the Authors' edits to be satisfactory to our previous points of concern.
Comments on the Quality of English Language
Please see the above section.
Author Response
Thank you very much for your comments and suggestions. As you can see, sections 2.1 and 2.2 have been removed from the body of the article, as they were originally. They have been moved to Supplementary Materials, as you suggested. Only minimal information about the content of these sections has been retained.
In addition, a repetition has been corrected (lines 69-70).
We are very pleased that you are satisfied with the changes that have been made. The article has been reviewed again and checked to ensure that it meets your requirements. As you have already seen, the entire document has been revised in order to improve its readability and comprehension. We hope that you are now fully satisfied with these changes.

Reviewer 2 Report
Comments and Suggestions for Authors
The authors took all the recommendations into account and significantly improved the manuscript. The article may be published in the journal "Life".
Author Response
The authors took all the recommendations into account and significantly improved the manuscript. The article may be published in the journal "Life".
-Thank you very much for your comments. We are delighted that our efforts to implement the improvements you suggested have resulted in such positive feedback from you.

Reviewer 3 Report
Comments and Suggestions for Authors
I maintain my previous comments; the paper is really too long, and therefore very difficult to follow. I also maintain that too much non-usefull information are provided. The paper is a mix between a review paper and a paper presenting new results. I led the editor make the decison regarding the lenght of the paper (about 1000 lines) and what could be removed or simplified.
Author Response
I maintain my previous comments; the paper is really too long, and therefore very difficult to follow. I also maintain that too much non-usefull information are provided. The paper is a mix between a review paper and a paper presenting new results. I led the editor make the decison regarding the lenght of the paper (about 1000 lines) and what could be removed or simplified.
-Thank you for your comments and suggestions. Sections 2.1 and 2.2 have been moved to Supplementary Materials. Only minimal information about the content of these sections has been retained.
